# Shift in the Light Quality of Night Interruption Affects Flowering and Morphogenesis of *Petunia hybrida*

**DOI:** 10.3390/plants12102049

**Published:** 2023-05-21

**Authors:** Yoo Gyeong Park, Byoung Ryong Jeong

**Affiliations:** 1Institute of Agriculture and Life Science, Gyeongsang National University, Jinju 52828, Republic of Korea; ygpark615@gmail.com; 2Division of Applied Life Science (BK21 Four), Graduate School, Gyeongsang National University, Jinju 52828, Republic of Korea; 3Research Institute of Life Science, Gyeongsang National University, Jinju 52828, Republic of Korea

**Keywords:** blooming, cryptochrome, photomorphogenesis, phytochrome, wavelength

## Abstract

*Petunia hybrida* Hort. “Easy Wave Pink”, a qualitative long-day plant (LDP), was investigated to study the effects of the night interruption light (NIL) provided by light-emitting diodes (LEDs) quality shifting on the morphogenesis, blooming, and transcription of photoreceptor genes. Plants were grown in a closed-type plant factory employing white (W) LEDs at an intensity of 180 μmol·m^−2^·s^−1^ PPFD provided for short day (SD, 10 h light, 14 h dark), long day (LD, 16 h light, 8 h dark), or SD with 4 h night interruption (NI) with LEDs at an intensity of 10 μmol·m^−2^·s^−1^ PPFD. The NIL quality was shifted from one light spectrum to another after the first 2 h of NI. Light treatments consisting of all possible pairings of W, far-red (Fr), red (R), and blue (B) light were tested. The SD and LD were referenced as the control, while 12 NI treatments involved altering LED NIL qualities, as follows: from R to B (NI-RB), from B to R (NI-BR), from Fr to R (NI-FrR), from R to Fr (NI-RFr), from Fr to B (NI-FrB), from B to Fr (NI-BFr), from B to W (NI-BW), from W to B (NI-WB), from W to Fr (NI-WFr), from Fr to W (NI-FrW), from W to R (NI-WR), and from R to W (NI-RW). The NI-RFr resulted in the longest shoots, while the NI-WR and NI-RW resulted in the shortest shoots. NI-WR, NI-RW, NI-BW, NI-WB, NI-RFr, NI-RB, NI-BR, and LD all exhibited flowering. High-level expressions of photoreceptor genes were confirmed in the NI-RFr, NI-FrR, NI-BFr, NI-RW, and NI-WR treatments. Morphogenesis and blooming were both impacted by the photoperiod. The first NIL had no effects on the flowering or the morphogenesis, but the second NIL had a profound impact on both.

## 1. Introduction

The light environment (photoperiod, light quality, light intensity, etc.) significantly affects a plant’s photomorphogenic development, including flowering, seed germination, and shoot architecture [1,2]. In photoperiodic plants, the phytochrome photoreceptors regulate perception of the light quality, stem extension, and flowering [3]. Plants identify the light quality via photoreceptors, which are categorized as phytochromes, cryptochromes, phototropins, members of the Zeitlupe (ZTL/FKF1/LKP2) family, and the ultraviolet (UV) light receptor(s) [4,5]. Phytochromes are photoreceptors that primarily absorb red (R) and far-red (Fr) light, while cryptochromes absorb UV-A and blue (B) light, both of which help plants bloom [6]. Phototropins play important roles in phototropism, changes in chlorophyll light-avoidance and accumulation movements, inhibition of rapid elongation of the hypocotyl growth, stomatal opening and closing, and leaf expansion [7].

Early flowering can be induced for commercial horticulture businesses by manipulating the photoperiodic conditions [8]. During short-day (SD) seasons, night interruption (NI) is successful in delaying short-day plants (SDPs) from flowering in similarly as to naturally long-day (LD) conditions do, and in hastening long-day plants (LDPs) to flower to allow for earlier sale or seed production [9]. A NI with a very low (3–5 μmol m^−2^ s^−1^ PPFD) light intensity promotes flowering induction and increases growth rates during the juvenile stage in *Cymbidium aloifolium* [10]. In *Arabidopsis thaliana*, an LDP, it was discovered that R light NI was the most effective at preventing flower blooming, and that this inhibition was frequently reversible with Fr light [11]. In *Petunia hybrida*, an LDP, NI treatments with green, Fr, and white (W) light encouraged flowering [12]. According to Shin et al. [13], NI treatment with a combination of B and R light encouraged *Cyclamen persicum* flowering in the winter. In *Pelargonium* × *hortorum*, a day-neutral plant (DNP), NI treatment with B, green, R, Fr, and W light encouraged flowering but delayed it in NI treatment with Fr light [14]. By keeping herbaceous SDPS in their vegetative growth stage, NI was also employed to prevent or delay flowering in *Dendranthema grandiflorum* [15,16] and *Kalanchoe blossfeldiana* [11]. *K. blossfeldiana* ’Lipstick’ (SDP) was not affected in flowering by any of the night interruption lights (NILs), such as B, R, W, or combination of B and W, while the ‘Spain’ variety flowered only in the 10 μmol m^−2^ s^−1^ PPFD NI-interruptional B light [17]. *D. grandiflorum* (SDP) responded the most strongly to R light NI for inhibiting flowering, but NI with Fr light or B light had less of an impact [18]. According to Yang et al. [19], 30 μmol m^−2^ s^−1^ PPFD supplemental B light and NI-interruptional B light were more effective in promoting growth, flowering, and the expression of florigen genes. Plant responses to light quality are species-specific, according to research conducted under various light environments [20].

However, NI using LEDs with varying light qualities was only tested on *D. grandiflorum* (SDP) [21] and *Pelargonium* × *hortorum* (DNP) [22] and not on LDPs. Our previous study on shifts in light quality of NI in SDP [21] and DNP [22] showed that morphogenesis and flowering were affected by the second NIL, but the first NIL had no effects on either. Flowering of SDP was observed in the NI-RB, NI-FrR, NI-BFr, NI-FrB, NI-WB, NI-FrW, NI-WFr, NI-WR, and SD treatments, and was especially promoted in the NI-BFr and NI-FrB treatments [21]. DNP was observed to flower in all NI treatments, and flowering was promoted in NI-RFr and NI-FrR treatments [22]. We have reported the shifting of NIL in SDP and DNP; our findings on LDP are also important. We hypothesized that the first and second NIL would affect morphogenesis and flowering in LDP. Furthermore, new practical applications of NIL quality shifting for the floricultural industry would be of substantial interest. The effects on *P*. *hybrida* Hort. “Easy Wave Pink”, an LDP, of NIL quality shifting on the blooming, morphogenesis, and transcription of the photoreceptor genes was therefore investigated in this work.

## 2. Results

### 2.1. Morphogenesis

NI-Fr resulted in the longest shoots, while NI-WR and NI-RW resulted in the shortest shoots (Figure 1). In comparison to that in LD, the leaf-length-to-leaf-width ratio dropped in the NI-WB, NI-BW, and NI-RW treatments (Figure 2A). The ratio of leaf length to petiole length grew in the NI-WB, NI-RB, NI-BW, and NI-WFr treatments as compared to the LD treatment (Figure 2B). In contrast, Fr light-driven decrease in the leaf-length-to-petiole-length ratio as seen in NI-FrW, NI-FrB, NI-BFr, NI-FrR, and NI-RFr, suggests succulent growth. NI-BR resulted in the greatest number of leaves per plant, followed by NI-WB and NI-RW (Figure 2C). The leaf area improved in all NI treatments and SD compared to that in LD, probably due to inhibition of flowering in petunia and continued vegetative growth in those deceptive photoperiods (Figure 2D).

The relative growth rate was the greatest in the NI-WB and the least in the SD treatments (Figure 3). With the exception of NI-BFr, all NI treatments led to higher chlorophyll content. The greatest chlorophyll content was found in NI-RB, NI-BR, NI-WB, and SD (Figure 4). The shoot fresh weight was the greatest in NI-BR, the shoot dry weight was the greatest in Ni-WB, and considerably higher biomass was obtained in all NI treatments (Table 1).

### 2.2. Flowering

The percentage of flowered buds observed was as follows: 100% in the NI-RW, NI-BR, and LD treatments; 75% in the NI-RFr and NI-WB treatments; and 50% in the NI-WR, NI-BW, and NI-RB treatments (Table 2 and Figure 5). The DVB increased in those NI treatments in which the plant flowered as compared to the LD (Table 2). Flowering results observed in this experiment are also a reflection of the R-to-Fr light ratio, and the effects of quality shifting on flowering was more pronounced by the second NIL than the first NIL.

High-level expressions of photoreceptor genes were confirmed in the NI-RFr, NI-FrR, NI-BFr, NI-RW, and NI-WR treatments (Figure 6). Compared to those in LD, phytochromes (*phyA* and *phyB*) were extensively expressed at elevated levels in the NI-RFr and NI-FrR treatments (Figure 6). *FTL* genes were studied under different light conditions during the NI along with the LD and SD, and compared to other conditions, the expression of *FTL* and *AFT* were more substantial in the NI-RFr and NI-FrR treatments.

## 3. Discussion

Numerous elements of plant development, including flowering, stem elongation, and seed germination are controlled by the plant hormone gibberellic acid [23] and light [24]. Increased shoot length observed in this study indicates the involvement of gibberellic acid as affected by the Fr light used in the NI. This study’s results show that NI-RFr caused the formation of the longest shoots, while NI-WR and NI-RW formed the shortest shoots, indicating that Fr light encouraged shoot extension while R and W lights inhibited it. Following exposure to Fr-rich light, many plant species greatly speed their elongation within minutes [23]. On the other hand, exposing plants to R-rich light again causes the extension to slow down to the same degree. Phytochromes are involved in such R/Fr light reversibility. Fr-rich light can accelerate flowering, reduce assimilate accumulation, reduce seed set, decrease fruit development, and impair seed quality, while increasing the elongation, which coincides with stronger apical dominance and reduced branching [25]. The increased shoot length observed in the NI-RFr treatment in this study might have been related to the effects of Fr light given in the second or last NIL. Furthermore, suppressed shoot length in both the NI-RW and NI-WR treatments might have been due to the relatively greater leaf expansion observed in the treatments with NIL composed of R and W lights.

The leaf-length-to-leaf-width-ratio decreased further in the NI-RW, NI-WB, and NI-BW treatments compared to that in the LD treatment, indicating relatively greater leaf expansion induced by the W light. The leaf area increased in all NI treatments and SD as compared to the LD treatment, probably due to inhibition of flowering in petunia and continued vegetative growth in those unreceptive photoperiods.

The greatest chlorophyll content was observed in SD, NI-RB, NI-BR, and NI-WB, which was probably enhanced by B light, especially given that, in the last NI period, the B light quickly and reversibly regulates the stomatal aperture, which enables a great stomatal aperture [26]. It is quite plausible that zeaxanthin [27], cryptochromes, and phototropins [28] are also involved in the B light signaling in the guard cells. Increased chlorophyll content in those treatments may have caused more active photosynthesis, resulting in enhanced biomass prompted by the NI even at low light intensities. 

In this study, the percentages of flowered buds observed were as follows: 100% in the NI-WR, NI-BR, and LD treatments; 75% in the NI-RFr and NI-WB treatments; and 50% in the NI-RW, NI-BW, and NI-RB treatments. Our previous study [12] suggested that percentage of flowering was greatest in LD (100%), followed by both NI-R (33.3%) and NI-Fr (33.3%) and both NI g (16.6%) and NI-W (16.6%) during SD with 4 h NI treatment. In our previous study, R, Fr, G, and W light induced flowering in petunia [12], but in this study with shift in 2 h NIL each, B light also affected flowering induction. This suggests that the second NIL quality has a greater effect on flowering induction than the first NIL quality due to the complexity of shift in NIL quality.

The phytochrome photoequilibrium, which affects flowering of photoperiodic crops, is influenced by the R-to-Fr light ratio (P_Fr_/P_R+Fr_) [3]. Flowering and stem extension in LDPs is promoted by a low P_Fr_/P_R+Fr_ [29]. Artificial lighting with Fr light, especially at the conclusion of the photoperiod, drives many LDPs to flower the most quickly [30,31]. *A. thaliana* flowered later when exposed continuously to high P_Fr_/P_R+Fr_ light than when grown with a low P_Fr_/P_R+Fr_ light [32]. Additionally, plants flowered much later when grown with continuous R light than when grown with continuous B light [33]. Flowering results of the petunia observed in this study are also the reflection of the P_Fr_/P_R+Fr_, and the effects of quality shifting on flowering was more pronounced by the second than the first NIL. In the model plant *A. thaliana*, the roles of *phyA* and *phyB* have been thoroughly explored [4]. It has been observed that plants under continuous Fr and R light with the *phyA* and *phyB* mutants grow taller than the wild type [34], and shows how *phyA* and *phyB* function to detect the appropriate wavelength of light to trigger the hypocotyl inhibitory response. In this study, the phytochromes (*phyA* and *phyB*) in petunia were extensively expressed at higher levels in the NI-RFr compared to in the LD. This implies that phytochromes (*phyA* and *phyB*) were involved in the Fr and R light perception for initiation of flowering in the petunia as also described previously in *A. thaliana* [35]. The roles of cryptochromes in inducing flowering have been observed by utilizing mutations in the *cry1* and *cry2* genes, on the other hand, because cryptochromes are known to stimulate flowering. In this study, the effects of the NIL quality, along with the LD and SD, were investigated, because it is well known that the LD and SD extensively affect flowering in *A. thaliana*. In the NI-RFr, NI-WR, and NI-RW treatments, since a high level of *cry1* gene expression was induced, the plants flowered, but not in the NI-FrR treatment. Flowering plants induced the expression of the *cry1* gene in NI-RFr, which explains that cryptochromes were receptors of these light conditions. In the NI-FrR treatment, even though a high level of *phyA*, *cry1*, and *FTL* gene expression was induced, the plants did not flower. In the NI-BFr treatment, even though *phyA* and *FTL* were highly expressed, the plants did not flower. Indeed, phenotypic studies also showed that the petunia had the receptors of the B, W, Fr, and R light even under photoperiods other than LD by showing blooming of flowers.

In addition, the genes similar to *FTL* and *TFL* play an important part in integrating endogenous and exogenous flowering-controlling signals [36]. The *FTL* and *TFL* encode small globular-like proteins. In this study, *FTL* genes were studied under different light conditions during the NI along with the LD and SD, and the expression of these genes (*FTL* and *AFT*) were more pronounced in NI-RFr, NI-FrR, and NI-BFr than in other conditions. This indicates that the *FTL* was also the receptor of the B, R, and Fr, in addition to the LD and SD. Overall, this study presumed that the B, Fr, and R light was mainly received by phytochromes, cryptochromes, and flowering terminal locus genes in the petunia.

## 4. Materials and Methods

### 4.1. Plant Materials and Growth Conditions

Petunia (Pan Seed Co., West Chicago, IL, USA) seedlings were transplanted from a glasshouse bench into 50-cell plug trays with a commercial medium (Tosilee Medium, Shinan Grow Co., Jinju, Republic of Korea) 40 days after sowing. On the day of transplanting, the seedlings and rooted cuttings were moved to a closed-type plant factory. After settling in for 24 days in the plant factory, the plants with shoot lengths of around 6.0 cm were exposed to the photoperiodic light treatments. For LDPs, the critical day length is 14 h, and for SDPs, it is 12 h. The plants were grown in a glass house and then moved to the plant factory, first to adapt to 20 ± 1 °C, 60 ± 10% RH, and 140 ± 20 μmol·m^−2^·s^−1^ PPFD from fluorescent lamps (F48T12-CW-VHO, Philips Co., Ltd., Eindhoven, The Netherlands) and subsequently acclimatize to the photoperiodic treatments provided by LED lighting systems at 25 cm atop the plant canopy. Throughout the experiment, the petunia was fertigated once every day with a greenhouse multipurpose nutrient solution [in mg·L^−1^ Ca(NO_3_)_2_·4H_2_O 737.0, KNO_3_ 343.4, KH_2_PO_4_ 163.2, K_2_SO_4_ 43.5, MgSO_4_·H_2_O 246.0, NH_4_NO_3_ 80.0, Fe-EDTA 15.0, H_3_BO_3_ 1.40, NaMoO_4_·2H_2_O 0.12, MnSO_4_·4H_2_O 2.10, and ZnSO_4_·7H_2_O 0.44]. 

### 4.2. Photoperiodic Light Treatments

The petunia was cultivated using white LEDs at an intensity of 180 μmol·m^−2^·s^−1^ PPFD for long day (16 h light, 8 h dark), short day (10 h light, 14 h dark), or SD with a 4 h (23:00–3:00) night interruption with LEDs at an intensity of 10 μmol·m^−2^·s^−1^ PPFD. The NIL quality was shifted from one to another after the first 2 h of NI until the end of the experiments for 66 days. The employed NIL qualities in this study were W (400–700 nm), R (660 nm), Fr (730 nm), and B (450 nm) (Figure 7). The SD and LD were referenced as the control in this study, and 12 NI treatments combined with different NIL combinations were studied, formulated as follows: B to R (NI-BR), R to B (NI-RB), R to Fr (NI-RFr), Fr to R (NI-FrR), B to Fr (NI-BFr), Fr to B (NI-FrB), W to B (NI-WB), B to W (NI-BW), Fr to W (NI-FrW), W to Fr (NI-WFr), R to W (NI-RW), and W to R (NI-WR) (Figure 8). A spectroradiometer (USB 2000 Fiber Optic Spectrometer, Ocean Optics Inc., Dunedin, FL, USA) scanned the spectral distribution of lights in all treatments 25 cm above the bench top in 1 nm intervals. In each light treatment, the average of the maximum absolute irradiance, and the spectral distribution were measured at three different locations within the plant-growing bench. 

### 4.3. Data Collection and Analysis

The leaf length, shoot length, petiole length, leaf width, average number of leaves, chlorophyll content, fresh and dry shoot and root weights, relative growth rate, percent flowering, days from the start of the photoperiodic treatments to the first visible flower bud or days to visible buds (DVB), average number of flowers, and expression of photoreceptor genes were all assessed after 66 days. The leaf expansion index was calculated as the proportion of the leaf length to leaf width, and the overgrowth (stretchiness) index was calculated as the proportion of the leaf length to the petiole length. The mean net increase in the dry biomass divided by the plant dry biomass over a period of time was taken as the relative growth rate. Before (W1) and after (W2) the treatments were applied, the total plant dry weight was measured, and the relative growth rate between finishing (t2) and starting (t1) days of the experiments was calculated as follows: Relative growth rate = (lnW2 − lnW1)/(t2 − t1)

In total, 10 mg of fresh, young completely formed leaves were extracted using 80% ice-cold acetone to estimate the chlorophyll content. The absorbance of the supernatant was assessed with a spectrophotometer (Biochrom Libra S22, Biochrom Co., Ltd., Holliston, MA, USA) at 663 and 645 nm after centrifugation at 3000 rpm. Dere et al. [37] was referenced for calculations. After drying for three days at 75 °C in an oven (Model FO-450M, Jeio Technology Co., Ltd., Seoul, Republic of Korea), the shoot and root dry weights were measured. 

However, treatment effects was also analyzed separately for the first (first NI) and second (second NI) 2 h periods of the 10 h short-day (SD) treatments based on the assumption that the group of the same light-quality treatments during the same period being the same treatment, e.g., in the first period the blue (B) treatment consisted of B to R (NI-BR), B to Fr (NI-BFr), and B to W (NI-BW).

This experiment used a randomized complete block design, with 3 replications, and 2 plants for each replication. To reduce the effects of the position, the treatment sites in a controlled setting were arbitrarily mixed between replications. The SAS (Statistical Analysis System, V. 9.1, Cary, NC, USA) program was used to determine the statistical significance of the acquired data. Duncan’s multiple range test and an analysis of variance (ANOVA) were applied to the results of this experiment. Graphing was completed using Sigma Plot 10.0 (Systat Software, Inc., San Jose, CA, USA).

### 4.4. Isolation of the Total RNA Isolation and Semi-Quantitative RT-PCR (Reverse Transcriptase- Polymerase Chain Reaction) Analysis of a Subset of Genes

Following the manufacturer’s protocols (Promega, Madison, WI, USA), the total RNA was isolated from the shoot tip of the plants that had been exposed to 33 days of NI treatments. Using a reverse transcriptase kit from Promega, Madison, WI, USA, 1 µg of DNase-treated RNA was reverse-transcribed to create first-strand cDNA, which was then utilized as the template for the PCR (polymerase chain reaction). The *phytochrome A* (*phyA*), *phytochrome B* (*phyB*), *cryptochrome 1* (*cry1*), *Anti-florigenic FT/TFL1* family protein (*AFT*), and *FLOWERING LOCUS T* (*FTL*) genes of the sequence from *Arabidopsis thaliana* were used as primers in separate PCRs with an equal amount of cDNA (Table 3). In petunia, similar to Arabidopsis, flowering is delayed under R light and induced under B light; however, its mechanism still remains unknown. Therefore, *A. thaliana* primers with similar gene sequences [38,39] were used. *Actin* was employed as the control because, due to its high conservation as an endogenous housekeeping gene, it is frequently used to normalize molecular expression investigations. The following PCR conditions were used: 5 min initial denaturation for at 95 °C, 35 20 s cycles at 95 °C, 30 s at 57 °C, 30 s at 72 °C, and a final extension step of 10 min at 72 °C. After 35 cycles, the PCR results were tested on a 1% agarose gel to determine whether the transcripts were expressed differently. 

## 5. Conclusions

NI-RFr resulted in the longest shoots, whereas NI-WR and NI-RW resulted in the shortest shoots, which indicates that FR light encouraged shoot extension. LD, NI-BR, NI-RB, NI-WR NI-WB, NI-RFr, NI-RW, and NI-BW caused plants to flower. Photoperiod affected both morphogenesis and flowering. While the first NIL had no effects on flowering or morphogenesis, the second NIL had a profound impact on both. 

## Figures and Tables

**Figure 1 plants-12-02049-f001:**
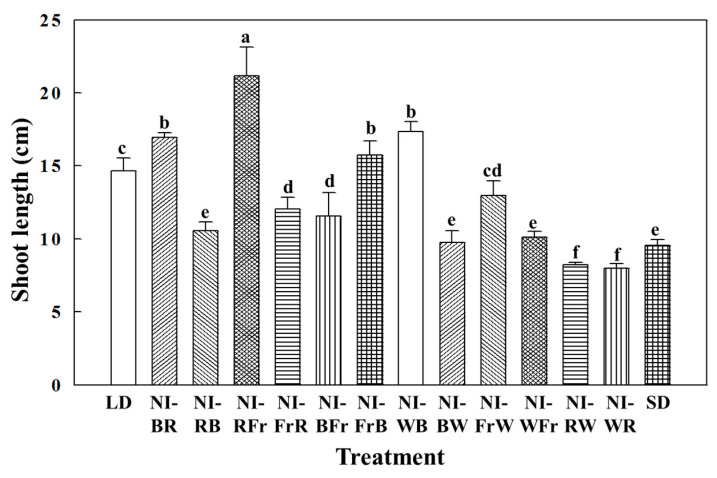
The effects of the NIL quality shifting at 10 μmol·m^−2^·s^−1^ PPFD on the shoot length of petunia (*Petunia hybrida* Hort. ‘Easy Wave Pink’), taken 66 days after treatment: NI-BR, blue to red; NI-RB, red to blue; NI-RFr, red to far-red; NI-FrR, far-red to red; NI-BFr, blue to far-red; NI-FrB, far-red to blue; NI-WB, white to blue; NI-BW, blue to white; NI-FrW, far-red to white; NI-WFr, white to far-red; NI-RW, red to white; and NI-WR, white to red. The LD indicates the 16 h long-day treatment. Vertical bars indicate means ± S.E. (*n* = 3). Means accompanied by different letters are significantly different (*p* < 0.05) according to the Duncan’s multiple range test at 5% significance level.

**Figure 2 plants-12-02049-f002:**
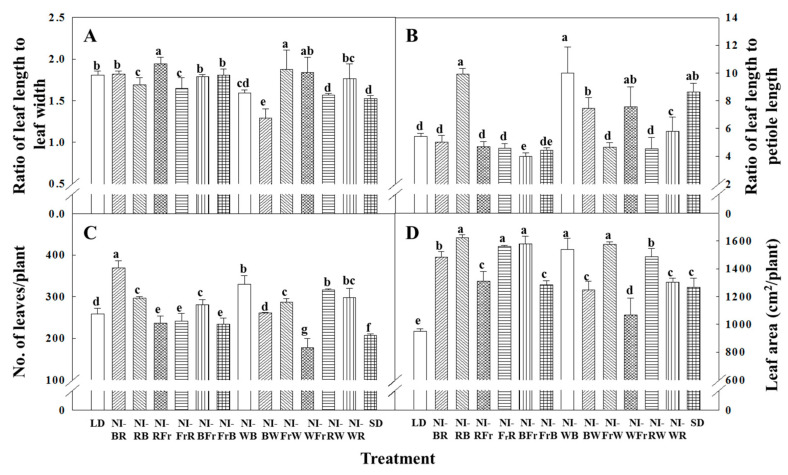
The effects of the NIL quality shifting at 10 μmol·m^−2^·s^−1^ PPFD on the leaf-length-to-leaf-width ratio (**A**), leaf length to petiole length ratio (**B**), average number of leaves (**C**), and leaf area (**D**) of petunia (*Petunia hybrida* Hort. ‘Easy Wave Pink’) taken 66 days after treatment. Please refer to Figure 1 for detailed NIL qualities. Vertical bars indicate means ± S.E. (n = 3). Means accompanied by different letters are significantly different (*p* < 0.05) according to the Duncan’s multiple range test at 5% significance level.

**Figure 3 plants-12-02049-f003:**
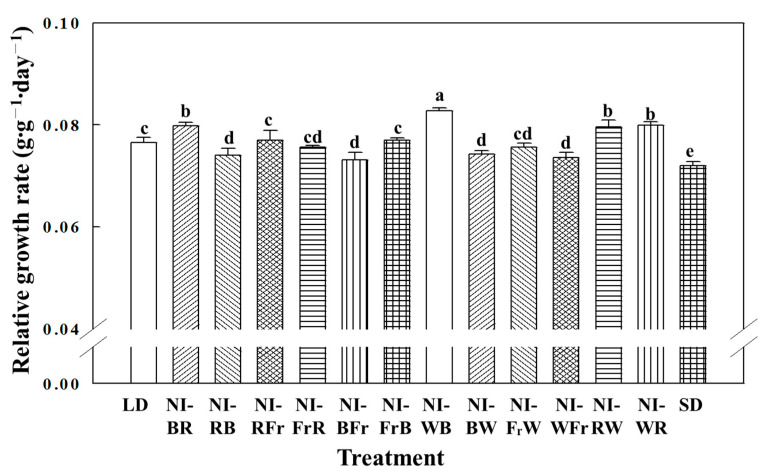
The effects of the NIL quality shifting at 10 μmol·m^−2^·s^−1^ PPFD on the relative growth rate of petunia (*Petunia hybrida* Hort. ‘Easy Wave Pink’) taken 66 days after treatment. Please refer to Figure 1 for detailed NIL qualities. Vertical bars indicate means ± S.E. (n = 3). Means accompanied by different letters are significantly different (*p* < 0.05) according to Duncan’s multiple range test at 5% significance level.

**Figure 4 plants-12-02049-f004:**
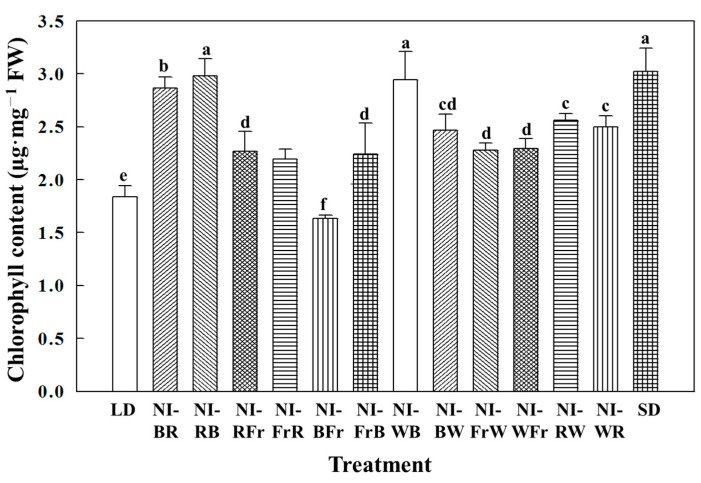
The effects of the NIL quality shifting at 10 μmol·m^−2^·s^−1^ PPFD on the chlorophyll content of petunia (*Petunia hybrida* Hort. ‘Easy Wave Pink’) leaves, taken 66 days after treatment. Please refer to Figure 1 for detailed NIL qualities. Vertical bars indicate means ± S.E. (n = 3). Means accompanied by different letters are significantly different (*p* < 0.05) according to Duncan’s multiple range test at 5% significance level.

**Figure 5 plants-12-02049-f005:**
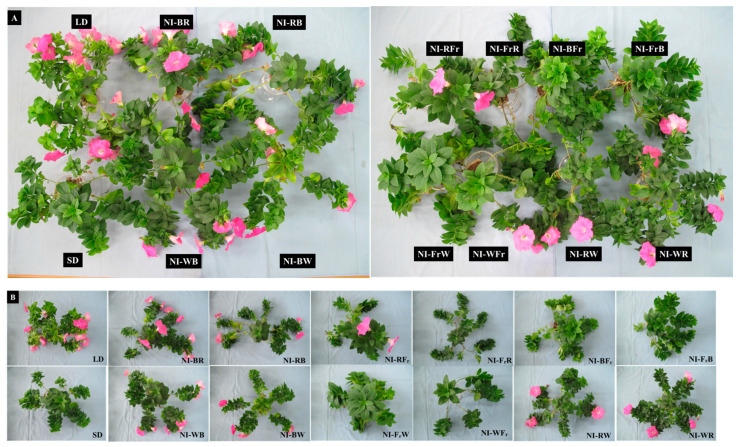
The effects of the NIL quality shifting at 10 μmol·m^−2^·s^−1^ PPFD on flowering of petunia (*Petunia hybrida* Hort. ‘Easy Wave Pink’), taken 66 days after treatment side view (**A**) and top view (**B**). Please refer to Figure 1 for the detailed NIL qualities.

**Figure 6 plants-12-02049-f006:**
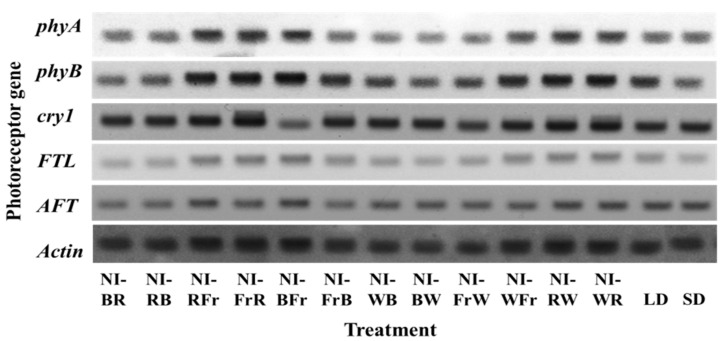
The effects of the NIL quality shifting at 10 μmol·m^−2^·s^−1^ PPFD on the expression of photoreceptor genes in petunia (*Petunia hybrida* Hort. ‘Easy Wave Pink’), taken 66 days after treatment. Please refer to Figure 1 for the detailed NIL qualities. The *phytochrome A* (*phyA*), *phytochrome B* (*phyB*), *cryptochrome 1* (*cry 1*), *Anti-florigenic FT/TFL1* family protein (*AFT*)*,* and *FLOWERING LOCUS T* (*FTL*) are all the indicated photoreceptor genes. Comparative gene expressions were conducted using a constitutively expressed *Actin* gene as the control.

**Figure 7 plants-12-02049-f007:**
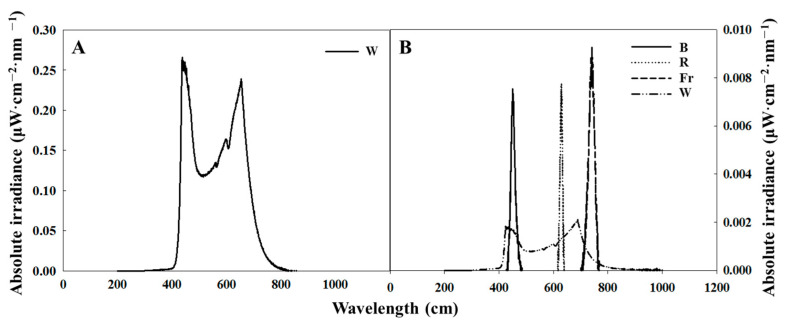
The spectral distribution of lights employed in the closed-type plant factory: White light-emitting diodes (LEDs) used as daily light (**A**) and different LEDs (B, blue; R, red; Fr, far-red; and W, white) used as night interruption light (NIL) provided by LEDs for 2 h each (**B**).

**Figure 8 plants-12-02049-f008:**
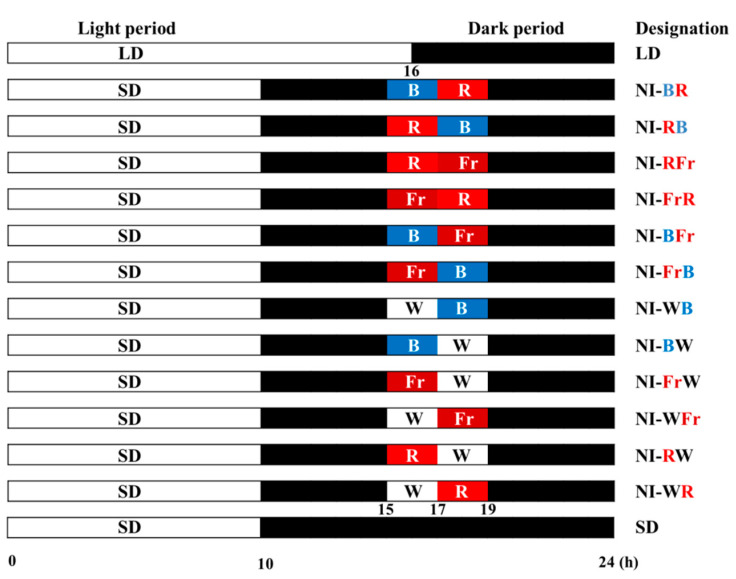
NIL quality shifting using LEDs during the 4 h night interruption (NI) in the 10 h short-day (SD) treatments. Please refer to Figure 1 for detailed NIL qualities.

**Table 1 plants-12-02049-t001:** The effects of the NIL quality shifting at 10 μmol·m^−2^·s^−1^ PPFD on the shoot and root fresh and dry weights in petunia (*Petunia hybrida* Hort. ‘Easy Wave Pink’), taken 66 days after treatment.

Treatment ^z^	Fresh Weight (g)			Dry Weight (g)		
	Shoot	Root	Total	Shoot	Root	Total
LD	64.8 bc ^y^	3.77 de	68.6 b–d	5.04 cd	0.35 e	5.37 cd
NI-BR	75.2 a	3.47 e	78.7 a	5.97 b	0.32 e	6.29 b
NI-RB	68.0 ab	4.96 c	73.0 a–c	4.45 d–f	0.36 de	4.82 c–e
NI-RFr	57.5 cd	4.35 c–e	61.9 de	5.08 cd	0.45 c	5.54 bc
NI-FrR	61.2 bc	5.82 b	67.0 b–d	4.57 d–f	0.59 b	5.16 c–e
NI-BFr	58.4 cd	6.11 ab	64.6 c–e	4.16 ef	0.46 c	4.63 de
NI-FrB	57.7 cd	6.67 a	64.4 c–e	4.85 de	0.67 a	5.52 bc
NI-WB	68.9 ab	6.73 a	75.6 ab	6.75 a	0.48 c	7.24 a
NI-BW	57.4 cd	4.68 c	62.1 de	4.51 d–f	0.34 e	4.85 c–e
NI-FrW	63.2 bc	3.67 de	66.8 b–d	4.85 de	0.32 e	5.17 c–e
NI-WFr	52.0 d	4.06 c–e	56.1 e	4.26 d–f	0.44 cd	4.70 c–e
NI-RW	64.7 bc	4.33 c–e	69.0 b–d	5.82 bc	0.43 cd	6.25 b
NI-WR	64.4 bc	4.53 cd	69.0 b–d	5.90 b	0.44 cd	6.34 b
SD	51.5 d	4.88 c	56.4 e	3.88 f	0.49 c	4.37 e
F-test	***	***	***	***	***	***

^z^ Please refer to Figure 1 for the detailed NIL qualities. ^y^ Mean separation within columns by Duncan’s multiple range test at a 5% level. ***: Significant at *p* ≤ 0.001, respectively.

**Table 2 plants-12-02049-t002:** The effects of the NIL quality shifting at 10 μmol·m^−2^·s^−1^ PPFD on the flowering characteristics of petunia (*Petunia hybrida* Hort. ‘Easy Wave Pink’), taken 66 days after treatment.

Treatment ^z^	Flowering (%)	DVB ^y^ (Day)	No. of Flowers/Plant
LD	100	31.0	49.6 a ^x^
NI-BR	100	62.6	20.3 b
NI-RB	50	35.0	4.0 cd
NI-RFr	75	59.6	4.3 cd
NI-FrR	- ^w^	-	-
NI-BFr	-	-	-
NI-FrB	-	-	-
NI-WB	75	73.6	7.3 c
NI-BW	50	50.6	4.3 cd
NI-FrW	-	-	-
NI-WFr	-	-	-
NI-RW	100	71.3	6.3 cd
NI-WR	50	50.6	3.6 cd
SD	-	-	-
F-test			NS ^v^

^z^ Please refer to Figure 1 for the detailed NIL qualities. ^y^ Days after treatment initiation to visible flower bud or days to visible buds. ^x^ Mean separation within columns by Duncan’s multiple range test at a 5% level. ^w^ No flowering. ^v^ NS: Nonsignificant.

**Table 3 plants-12-02049-t003:** Primers used to quantify gene expression levels.

Gene	Accession No.	Forward Primer	Reverse Primer
*phyA*	EU915082	5′-GACAGTGTCAGGCTTCAACAAG-3′	5′-ACCACCAGTGTGTGTTATCCTG-3′
*phyB*	NM_127435	5′-GTGCTAGGGAGATTACGCTTTC-3′	5′-CCAGCTTCTGAGACTGAACAGA-3′
*cry1*	NM_116961	5′-CGTAAGGGATCACCGAGTAAAG-3′	5′-CTTTTAGGTGGGAGTTGTGGAG-3′
*AFT*	AB839766	5′-AGAACACCTCCATTGGATCG-3′	5′-CTGGAACTAGGTGGCCTCAC-3′
*FTL*	AB839767	5′-ACAACGGACTCCTCATTTGG-3′	5′-CGCGAAACTACGAGTGTTGA-3′
*Actin*	AB205087	5′-CGTTTGGATCTTGCTGGTCG-3′	5′-CAGGACATCTGAAACGCTCA-3′

## Data Availability

Data sharing is not applicable to this article.

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
