# Peer review of "Shift in the Light Quality of Night Interruption Affects Flowering and Morphogenesis of Petunia hybrida"

_plants, 2023, doi:10.3390/plants12102049_

Round 1

Reviewer 1 Report

The possible effects of different night-interrupting light combinations are discussed in the MS by Park et al. The MS is interesting, but has many flaws.

The abstract is not informative and very difficult to read. I was able to understand it after reading the whole text several times. It should be rewritten more consistently.

The introduction is also confusing and very short. Since the photoreceptors play the main role during the night interruption, the whole MS should focus on them and not on emphasizing the physiological parameters.

The graphs and tables are informative, but I miss the statistical analysis of the vertical bars.

The MS part should also be completed, I could not find any information about the length of the total illumination period. Did it end at the first flower buds or did it continue until the end of the experiments (66 days)? It is also strange why there were two different SD treatments (was it 10 or 12 hours for SD treatments). In 4.2 it is written that the red LEDs were at 660 nm, but in Figure 7 it showed that the red LEDs have a maximum at around 630 nm, which is true? I also could not find any information about what is t1 and t2 at the determination of the relative growth (I think these are the starting and finishing days of the experiments, but not sure). And one main question at the end is why Arabidopsis genes were used for primers for Petunia. I know that the Petunia genome is not available, but using a closer relative (especially from the same Solanaceae family, like tomato) would be a better choice. And finally, I haven't seen a PCR experiment like this published in the last decade, rtPCR gives much more information with the appropriate primers.

I think the whole MS should be rethought, rewritten according to the fact of what happens regarding the photoreceptors under the different wavelengths, and conclude the results again with the results already presented to the photoreceptors.

Author Response

Thank you for review this manuscript.

The abstract is not informative and very difficult to read. I was able to understand it after reading the whole text several times. It should be rewritten more consistently.

Response> Authors revised abstract section.

The introduction is also confusing and very short. Since the photoreceptors play the main role during the night interruption, the whole MS should focus on them and not on emphasizing the physiological parameters.

Response> Authors revised introduction section.

The graphs and tables are informative, but I miss the statistical analysis of the vertical bars.

Response> Authors added the statistical analysis of the vertical bars.

The MS part should also be completed, I could not find any information about the length of the total illumination period. Did it end at the first flower buds or did it continue until the end of the experiments (66 days)?

Response> Illumination period continued until the end of the experiments. Authors added illumination period in materials and methods section.

It is also strange why there were two different SD treatments (was it 10 or 12 hours for SD treatments).

Response> We was carried out only 10 hours for SD treatments. SD with NI was 12 treatments.

In 4.2 it is written that the red LEDs were at 660 nm, but in Figure 7 it showed that the red LEDs have a maximum at around 630 nm, which is true?

Response> It may look like at around 630 nm on the graph, but a maximum at around 660 nm is correct.

 I also could not find any information about what is t1 and t2 at the determination of the relative growth (I think these are the starting and finishing days of the experiments, but not sure).

Response> Authors added t1 and t2 at the determination of the relative growth.

And one main question at the end is why Arabidopsis genes were used for primers for Petunia. I know that the Petunia genome is not available, but using a closer relative (especially from the same Solanaceae family, like tomato) would be a better choice. And finally, I haven't seen a PCR experiment like this published in the last decade, rtPCR gives much more information with the appropriate primers.

Response> In this study, we simultaneously conducted a study on the shift in NIL quality of petunia, geranium, and chrysanthemum to study the effects on morphogenesis and flowering, and each manuscript was published separately due to the large volume. Therefore, we used the Arabidopsis gene, which is often used as a model plant. As the reviewer said, it would have been nice to use the tomato gene, which is a member of the eggplant family, but it is too bad. This study has already been completed, and if we conduct similar studies in the future, we will reflect this.

I think the whole MS should be rethought, rewritten according to the fact of what happens regarding the photoreceptors under the different wavelengths, and conclude the results again with the results already presented to the photoreceptors.

Response> Authors was rewritten according to the fact of what happens regarding the photoreceptors under the different wavelengths, and conclude the results again with the results already presented to the photoreceptors.

Reviewer 2 Report

Dear Author, 

well done the good work, few spelling mistakes and the units of the measurement should be standardize. 

Author Response

Thank you for review this manuscript.

We revised introduction and discussion.

Reviewer 3 Report

This manuscript investigates the effects of night interruption light (NIL) provided by light emitting diodes (LEDs) quality shifting on the morphogenesis, blooming, and transcription of photoreceptor genes in Petunia hybrida Hort. “Easy Wave Pink”, a qualitative long day plant (LDP). The study found that Fr light encouraged shoot extension while R and W lights inhibited it. Furthermore, increased chlorophyll content observed in certain treatments may have caused more active photosynthesis resulting in enhanced biomass prompted by the NIL even at low light intensities. Flowering results observed were also a reflection of the R to Fr light ratio with effects of quality shifting on flowering being more pronounced by the second than the first NIL. Finally, phytochromes (phyA and phyB), cryptochromes and flowering terminal locus genes were presumed to be involved in receiving Fr and R light signals for initiation of flowering in petunia under different photoperiodic conditions. Overall, this manuscript provides valuable insights into how LED lighting can be used to manipulate plant growth and development for commercial horticulture businesses.

1. Line 37: The word "hastening" should be corrected to "hasten".

2. Line 44: The phrase "reversible with reversible with far red (Fr) light" is repeated twice, one of them needs to be deleted.

3. Line 53-54: It would be useful if we could explain why only SDP used LEDs with different emission qualities for testing.

4. Why was a PPFD of 10μmol·m-2·s-1 chosen for night interruption in the Materials and Methods section?

5. Lines 53-54: However, NI using LEDs with varying light qualities was only tried on SDPs  [12] and not on LDPs.

I note that literature 12 was published in 2015. The author needs to refer extensively to the literature in recent years to simply state what has been done.

6 how about the application effect of NIL in plants? It should be explained in the Introduction.

Author Response

Thank you for review.

  1. Line 37: The word "hastening" should be corrected to "hasten".

Response> Author's revised based on reviewer's comments.

  1. Line 44: The phrase "reversible with reversible with far red (Fr) light" is repeated twice, one of them needs to be deleted.

Response> Author's revised based on reviewer's comments.

  1. Line 53-54: It would be useful if we could explain why only SDP used LEDs with different emission qualities for testing.

Response> In this study, we simultaneously conducted a study on the shift in NIL quality of petunia, geranium (published), and chrysanthemum (published) to study the effects on morphogenesis and flowering, and each manuscript was submitted separately due to the large volume.

  1. Why was a PPFD of 10μmol·m-2·s-1 chosen for night interruption in the Materials and Methods section?

Response> As shown in reference 21, we conducted the experiment at a low light level because of the effects on morphogenesis and flowering at low light levels. 10 μmol·m-2·s-1 is a little higher than that light intensity, so we wanted to make sure that the results of the night treatment were clear.

  1. Lines 53-54: However, NI using LEDs with varying light qualities was only tried on SDPs [12] and not on LDPs.

Response> Authors revised it.

I note that literature 12 was published in 2015. The author needs to refer extensively to the literature in recent years to simply state what has been done.

Response> Authors added the literature in recent years.

6 how about the application effect of NIL in plants? It should be explained in the Introduction.

Response> Authors added the effect of NIL in plants.

Round 2

Reviewer 1 Report

The MS has improved since the last version, but I still don't understand why the authors chose the Arabidopsis genome for their experiments. I don't think it's any more suitable for geranium and chrysanthemum than it is for petunia. All three flowering plants are from different families, far from Arabidopsis, which is included in the Brassicaceae family. It would be better to run the experiments separately and at least check the results of the PCR experiment by sequencing the products to know what the result was. It is possible that the primers were able to target the same genes as in Arabidopsis, but no one knows it without further analysis of the products.

Author Response

I completely agree with the reviewer, and I think this is a disappointing aspect of the study. However, the study was completed a long time ago, and the samples were removed, so it is difficult to analyze them further. I hope that you will consider these aspects when judging this manuscript. Thank you.